# A Q-Learning-Based Load Balancing Method for Real-Time Task Processing in Edge-Cloud Networks

Zhaoyang Du [1], Chunrong Peng [2], Tsutomu Yoshinaga [1] and Celimuge Wu [1,*]

[1] Department of Computer and Network Engineering, The University of Electro-Communications, Tokyo 1828585, Japan; duzhaoyang@uec.ac.jp (Z.D.); yoshinaga@uec.ac.jp (T.Y.)
[2] Inner Mongolia University of Finance and Economics Library, Inner Mongolia University of Finance and Economics, Hohhot 010051, China; pengchunrong1978@gmail.com
[*] Correspondence: celimuge@uec.ac.jp

**Abstract:** Edge computing has emerged as a promising solution to reduce communication delays and enhance the performance of real-time applications. However, due to the limited processing power of edge servers, it is challenging to achieve efficient task processing. In this paper, we propose a Q-learning-based load-balancing method that optimizes the distribution of real-time tasks between edge servers and cloud servers to reduce processing time. The proposed method is dynamic and adaptive, taking into account the constantly changing network status and server usage. To evaluate the effectiveness of the proposed load-balancing method, extensive simulations are conducted in an Edge-Cloud network environment. The simulation results demonstrate that the proposed method significantly reduces processing time compared to traditional static load-balancing methods. The Q-learning algorithm enables the load-balancing system to dynamically learn the optimal decision-making strategy to allocate tasks to the most appropriate server. Overall, the proposed Q-learning-based load-balancing method provides a dynamic and efficient solution to balance the workload between edge servers and cloud servers. The proposed method effectively achieves real-time task processing in edge computing environments and can contribute to the development of high-performance edge computing systems.

**Keywords:** cloud computing; edge computing; Q-learning; load balancing





## 1. Introduction

With the rapid development and widespread application of IoT and 5G technologies, the growth of smart mobile devices and applications is exploding, rapidly increasing network traffic and service data volume, as well as users' demand for data processing speed and quality service experience [1–3]. Cloud computing could provide computing and storage services for data generated by smart mobile devices, breaking through the limited resources of terminal devices [4,5]. However, the long communication link between the cloud data center and users causes a larger service delay. The rise of edge computing provides reliable technical support for IoT, being closer to the data source [6,7]. It places latency-sensitive applications or tasks on the edge nodes for computing without going through longer links to upload data to the cloud server. This not only enables fast responses for critical tasks but also effectively relieves the computational pressure on the cloud data center.

Currently, the cloud-edge joint computing paradigm has gradually replaced the traditional computing paradigm and provided new momentum for the new Internet infrastructure [8,9]. Based on the collaboration between the cloud data center and the edge computing center regarding resources, networks, applications, and other aspects, the computing and processing at the edge release the pressure on the cloud data center to a certain extent, forming a more powerful cloud-edge joint data center. The rapid development of the Internet industry and information technology in recent years has led to an alarming expansion

of data volume on the Internet, drawing focused attention to the quality of service and resource utilization efficiency of the joint cloud-edge system. The task deployment method of the cloud-edge joint system directly determines the user service quality and system resource utilization efficiency. Therefore, many researchers are concerned about adopting reasonable and efficient deployment strategies and methods to achieve the highest quality service capability and resource utilization efficiency [10,11].

The motivation for this study is the escalating demand for real-time applications and the necessity for streamlined task processing in Edge-Cloud networks. Edge computing has emerged as a promising approach to processing data in proximity to the source, thereby mitigating communication delays and enhancing the performance of real-time applications. Nevertheless, the limited processing power of edge servers poses a significant challenge in achieving efficient task processing. Conventional static load-balancing methods prove inadequate in addressing the dynamic and unpredictable nature of Edge-Cloud networks. Consequently, a dynamic and adaptive load-balancing approach is imperative, capable of optimizing the allocation of real-time tasks between edge servers and cloud servers, thereby reducing processing time. The main contributions of the paper are listed below:

- We propose a load-balancing method based on reinforcement learning between edge servers and cloud servers to reduce the average service time. Service time refers to a task sent from the mobile device to the server and returns to the mobile device after the task is processed on the server side. In other words, the service time is the sum of the task processing time and the communication time.
- We propose a load-balancing method that could flexibly respond to network and server state changes using Q-learning in reinforcement learning. Moreover, in order to utilize edge computing to reduce communication delay, the proposed load-balancing method also considers the real-time tasks that need to be processed.
- We evaluate the proposed method and several other existing load-balancing methods through simulation and verify its superiority.

The remainder of the paper is organized as follows. Related works are listed in Section 2. The proposed Q-learning-based load-balancing method between cloud servers and edge servers is introduced in Section 3. Then, we present the performance of the proposed in Section 4. Finally, we conclude our work in Section 5.

## 2. Related Work

### 2.1. Load Balancing Method in Cloud-Edge-Client Communications

Cloud computing could reduce the deployment and operation costs of network services. However, when massive amounts of data are uploaded to the cloud servers for computing, it may lead to an unbalanced load among the cloud servers, thereby degrading the computing performance. Moreover, the communication link between the cloud servers and the user is too long, and the communication process is affected by factors such as network bandwidth in the data transmission process, the service delay is large, and it cannot provide high-quality and low-latency services [12].

The development of edge computing technology has alleviated the above problems. Deploying servers at the network edge could provide users with low-latency and high-efficiency services [13]. Delay-sensitive applications or task requests do not need to be uploaded to the cloud server through a long link so that the computing results can be obtained quickly. The emergence of edge computing not only meets the user's demand for delay but also relieves the computing pressure of cloud servers.

With the rapid development of the Internet industry and information technology, the amount of data on the Internet is increasing at an alarming rate, and a large number of computing tasks and applications are emerging [14]. The quality of service and resource utilization efficiency of the cloud-edge joint system have received significant attention.

In the Cloud-Edge-Client computing framework, devices can offload computing tasks to edge servers, execute tasks at the edge server, and return results [15]. The computing offloading process can be divided into two steps. One is to determine the target computing

server for computing offloading, and the other is to allocate communication and computing resources involved in the task data transmission process. In this process, the delay of tasks mainly includes the transmission delay of task data and the processing delay of computing tasks. To ensure system performance, how to perform task offload scheduling and network resource allocation to optimize the entire network operation is a significant problem to be solved. An efficient task offloading strategy can significantly improve the service quality and stability of the Cloud-Edge-Client computing framework.

Mahmoodi et al. propose cloud offloading and scheduling using dependency graphs of mobile application components [16]. Optimal co-scheduling of parallel processing elements between devices and cloud servers reduces energy consumption. Chiang et al. propose a Cotask-aware offloading scheme to leverage parallel processing on devices and edges to speed up computation [17]. Linear Programming Rounding (LPR) and Earliest Cotask Arrival First (ECAF) method are proposed for reducing the average Cotask completion time. Li et al. propose an architecture to support a computing structure combining cloud computing and edge computing from software and hardware, respectively, [18]. On the software side, distributed deep learning based on edge computing is used. On the hardware side, compression coding technology is used to reduce communication costs and increase system response speed. Ref. [19] proposed a method for AI-optimized workload allocation at the Cloud-Edge-Client computing framework to achieve the lowest response time for delay-sensitive applications. The proposed approach has two scenarios, a single workload scenario and a multi-job scenario, which are efficient allocation algorithms to reduce end-to-end response time. Several simulations verify that the end-to-end response time of the multi-job scenario outperforms other baseline methods.

### 2.2. Load Balancing Method Using Reinforcement Learning

Reinforcement learning is a type of machine learning that deals with the problem of observing the current state of an agent in an environment and deciding what actions to take. Agents acquire rewards from the environment by selecting actions and learning the policy that gives the most rewards through a series of actions. Typical reinforcement learning methods include Q-learning, the Monte Carlo method, and DQN (Deep Q Network) [20–22].

Ref. [23] modeled the computing offloading problem in the MEC (mobile edge computing) environment as a Markov decision process model and used the state transition process of the Markov chain to find the optimal solution. Ref. [24] considered the number of computing tasks being executed based on Ref. [23], which expanded the system state space and possible event space, thus enhancing decision-making accuracy. Ref. [25] proposed a deep Q-learning network to schedule tasks according to the energy consumption during task processing, which reduces system energy consumption and improves service quality. Ref. [26] proposed a non-model deep reinforcement learning framework under the MEC framework of heterogeneous cellular networks and configures the framework's parameters to allocate bandwidth more reasonably and save energy consumption, improving the utilization rate of the system. Ref. [27] explored the exciting fusion of wireless communications and multi-access edge computing beyond fifth-generation networks. The authors delved into the computation offloading problem and propose a distributed learning framework to tackle the challenges arising from uncertainties and limited resources. Ref. [28] presented a novel routing scheme that utilizes reinforcement learning and client-Edge-Cloud collaboration to efficiently create and adapt communication routes in a proactive and preemptive manner.

The disadvantage of several schemes using deep reinforcement learning networks is that the introduction of neural network model training will increase the completion time of computing tasks. For heterogeneous IoT networks, neural networks usually require a longer training time.

## 3. Proposed Load Balancing Method

### 3.1. Methodology Section

The proposed research aims to evaluate the effectiveness of a Q-learning-based load-balancing method for edge-cloud networks by conducting extensive simulations using EdgeCloudSim [29], a simulation tool designed for edge computing scenarios. Adopting a quantitative approach, the study seeks to assess the efficacy of the proposed load-balancing method through performance metrics such as average service time, task failure rate, and resource utilization efficiency. Simulations are utilized to generate data under diverse workload conditions, enabling a comparison of the proposed method's performance against existing baseline techniques.

We generated a set of real-time tasks with different processing requirements and arrival rates, and then simulated their processing on both edge servers and cloud servers under different network conditions.

In our study, we conduct a performance evaluation of our proposed Q-learning-based load-balancing method, comparing it against three traditional static load-balancing methods. We adopt a simulation-based approach to generate a substantial number of real-time tasks with diverse processing requirements and arrival rates. Subsequently, we simulate their processing on both edge servers and cloud servers under varying network conditions. First, we compute the average service time and task failure rate for each method, considering different network conditions, including varying WAN and LAN bandwidths. Next, we proceed to compare the performance of our proposed method with the other methods, revealing that our approach notably reduces the average service time and task failure rate, particularly when the network bandwidth is limited. The outcome of our result analysis presents compelling evidence for the effectiveness of our proposed Q-learning-based load-balancing method, demonstrating its superiority over traditional static load-balancing methods in edge-cloud networks.

In conclusion, the proposed research employs a quantitative approach to evaluate the effectiveness of a Q-learning-based load-balancing method for edge-cloud networks. By conducting simulations to generate data under diverse workload conditions and comparing the proposed method's performance against established baseline techniques, we ensure the validity, reliability, and generalizability of our findings. Applying purposive sampling aids in the focused selection of workload conditions, while statistical analysis guarantees rigorous examination of performance metrics. Implementing the Q-learning algorithm in Python, guided by the Q-value update rule, further adds robustness to the study. This comprehensive methodology offers valuable insights for the development of high-performance edge computing systems.

### 3.2. The Proposed Q-Learning Approach

Cloud computing can provide large-scale computing resources; however, the communication delay between the cloud server and clients cannot be ignored. Edge computing is introduced to reduce the communication delay, but the processing power of edge servers is inferior to that of cloud servers. To make use of the advantages of cloud computing and edge computing, a load-balancing method is performed between cloud servers and edge servers.

In this paper, we propose a load-balancing method using reinforcement learning between edge servers and cloud servers for the purpose of reducing average service time. Service time is the sum of the time required for task processing and communication time between cloud servers and edge servers. We aim to realize efficient load-balancing that can deal with all situations using Q-learning, which is a kind of reinforcement learning. Moreover, in order to take advantage of edge computing, which reduces communication delays, the proposed method attempts load-balancing by considering real-time tasks.

Specifically, the proposed method utilizes a Q-learning algorithm to determine the optimal action to take for a given state. The state is defined by three parameters: the data size of the task, the processing capacity of the edge server, and the real-time requirement

of the task. The real-time requirement of the task is evaluated in two levels, with level 0 denoting a task that does not require real-time processing and level 1 representing a task that requires real-time processing.

The action comprises two options: processing the task on the edge server or offloading the task from the edge server to the cloud server for processing. As shown in Table 1, when choosing action (1), tasks can be sent to the edge server with minimal communication delay. However, when selecting action (2), the task is sent to the cloud server, and communication between devices and the cloud server takes a significant amount of time. Thus, a decision on which action to perform should consider these factors.

The reward is defined as the reduction in processing time achieved by selecting a particular action. The proposed method uses the Q-learning algorithm to learn the optimal action to take for a given state by maximizing the expected reward over time. By continuously learning and adapting to the changing network conditions and server usage, the proposed method can achieve efficient task processing and enhance the performance of real-time applications.

The state s, action a, reward r, and the setting of the Q-value are important when using Q-learning. We define the proposed Q-learning method in Table 1. The Q-learning algorithm is a type of reinforcement learning that involves learning an optimal policy for an agent to take actions in an environment to maximize a reward signal. In the context of load-balancing, the agent is the load balancer, and the environment is the Edge-Cloud network.

The Q-learning algorithm updates the Q-value of a state-action pair based on the reward received and the maximum Q-value of the next state. The Q-value represents the expected cumulative reward of taking a particular action in a given state.

In Table 1, the state is defined by two parameters: the CPU usage rate of the edge server and the cloud server, and the task size. The action is defined as either processing the task on the edge server or offloading the task to the cloud server. The reward is defined as the reduction in processing time achieved by selecting a particular action.

The Q-value is updated using the Q-learning equation, which is a recursive formula that updates the Q-value of a state-action pair based on the reward received and the maximum Q-value of the next state. $Q(S_t, a_t)$ is the Q-value of the current state-action pair, $\alpha$ is the learning rate, r is the reward received for taking the action a in state s, $\gamma$ is the discount factor that determines the importance of future rewards, $\max a_{t+1} Q(S_{t+1}, a_{t+1})$ is the maximum Q-value of the next state-action pair, and $S_{t+1}$ and $a_{t+1}$ are the next state and action, respectively.

**Table 1.** Setting of each parameter in Q-learning.

| |
|---|
| State S: (1) CPU usage rate (Edge server/Cloud Server)<br>(2) Task size<br>(3) Delay sensitivity level 0 or 1 |
| Action A: (1) Task processed at Edge Server<br>(2) Task offloaded to cloud server |
| Reward r: Service time (Task processing time + communication delay)<br>R: (1) Task processing failure: R(1) = 0<br>(2) Task processing success (service time longer than average): R(2) = 10<br>(3) Task processing success (service time less than average): R(3) = 100<br>(4) Tasks that require real-time processing are processed by edge servers: R(4) = 50 |
| Value Q: $Q(S_t, a_t) \leftarrow (1 - \alpha)Q(S_t, a_t) + \alpha \left\{ \left( \frac{1}{r_{t+1}} + R \right) + \gamma \max_{a_{t+1}} Q(S_{t+1}, a_{t+1}) \right\}$ |

### 3.2.1. State S

We define three states (1) to (3) for State S. State (1) represents the CPU usage rate of each edge server and cloud server, which is a crucial factor in load-balancing and greatly influences offload decisions. State (2) represents the size of the task to be processed. Task size directly affects processing time and communication time, making it important to

decide which server should handle the task. State (3) evaluates the real-time requirement of the task in two levels from 0 to 1. Level 0 denotes a task that does not require real-time processing and does not need to consider communication delays. In contrast, level 1 represents a task that requires real-time processing, and processing them on edge servers with low communication delays is preferable. The presence or absence of this real-time requirement is critical in determining how to distribute the load in the proposed scheme.

### 3.2.2. Action A

Action A comprises two options: (1) processing the task on the edge server or (2) offloading the task from the edge server to the cloud server for processing. When choosing option (1), tasks can be sent to the edge server with minimal communication delay. However, when selecting option (2), the task is sent to the cloud server, and communication between devices and the cloud server takes a significant amount of time. Thus, a decision on which action to perform should consider these factors.

### 3.2.3. Reward r and R

We define two types of rewards, namely, r and R. Reward r represents the service time of each task. As the service time remains constant, it does not significantly impact the updating of the Q-value. Hence, the reward R becomes more critical in the load-balancing process. The reward R is determined based on whether the task processing failed or succeeded. A reward of (1) is given if the task processing fails. On the other hand, if the task has been successfully processed, a reward of (2) is given if the service time of the processed task is longer than the average service time at that time, and (3) is given if it is shorter. Since successfully processing the task and reducing the average service time is the most desirable reward, the magnitude of the reward increases in the order of (1)→(2)→(3).

In addition to rewards (1) to (3), a reward (4) is also given when an edge server processes a level 1 task that requires real-time processing. Consequently, the highest reward of (3)+(4) is awarded when the edge server successfully processes the level 1 task and the service time is less than the average service time.

### 3.2.4. The Update of Q Value

Finally, we introduce the equation for updating the Q-value. The proposed method is based on the general Q-learning equation, but only the parts related to the reward are modified. The reward r is directly proportional to the inverse of the service time, and the reward R is proportional to the magnitude of the reward given according to the task processing result. Thus, a smaller service time and a larger reward value will result in a larger Q-value. Therefore, the Q-value will gradually increase as the learning process is repeated, enabling it to respond optimally in various situations. The Q-value is updated each time a task is processed, and an increase in the Q-value indicates a more efficient load-balancing method.

## 4. Simulation Result and Analysis

### 4.1. Simulation Tool and Setup

To verify the performance of the proposed method against existing baselines, we utilized EdgeCloudSim [29], an open-source tool based on CloudSim that provides a simulation environment tailored to edge computing scenarios. The present study introduces a novel Q-learning-based load-balancing approach, which is implemented within the framework of an edge orchestrator module, as depicted in Figure 1. This module serves the purpose of load allocation between cloud servers and edge servers. EdgeCloudSim enables experiments to be conducted considering both computing and networking resources, allowing for a comprehensive evaluation of the proposed method.

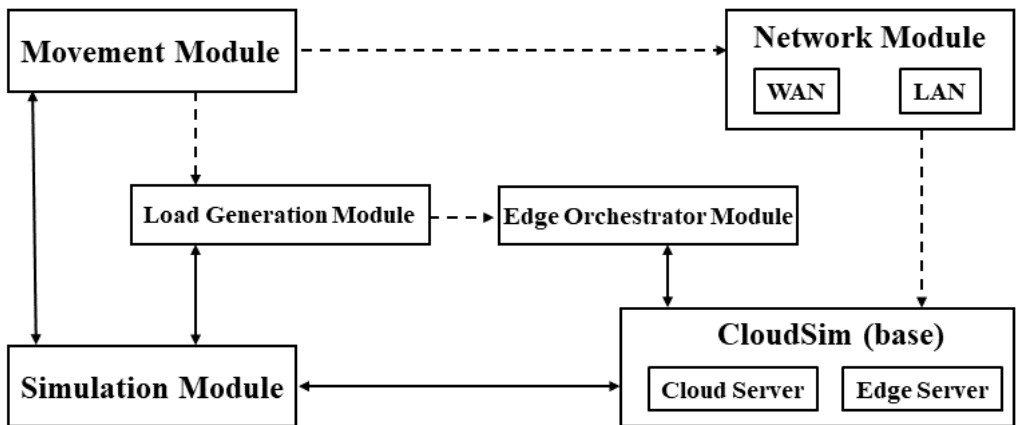

**Figure 1.** The relationship between EdgeCloudSim modules.

In the simulation scenario, we modeled a college campus where students move around and request services from edge servers located in specific buildings. The services consist of four applications, with each student's device executing one application based on a predefined usage rate. The students followed a nomadic mobility model, where they moved to another location after a random period. This scenario enabled us to evaluate the performance of the proposed method in a realistic and dynamic edge computing environment.

The communication environment used in the simulation is illustrated in Figure 2. The wireless LAN (Local Area Network) connects the devices to the edge server, while the devices are connected to the cloud server via WAN (Wide Area Network). The simulation includes multiple edge servers, and the servers located in different areas are connected by the Metropolitan Area Network (MAN). However, this paper does not consider the load-balancing among the edge servers, so it does not significantly impact the simulation results.

The parameters for the cloud server and edge servers are presented in Table 2 and Table 3, respectively. To reduce the service time, it is ideal to place a server with high processing power near the user. However, from the perspective of server operation and setup costs, providing such high processing power to all users is not feasible. Therefore, in this paper, only the cloud server is equipped with a powerful processing capacity, while the edge servers are designed with low processing capacity. Although there are a total of 14 edge servers, load balancing among them is not considered in our study.

The simulation parameters regarding time and communication scale are presented in Table 4. These parameters are set as the initial values for the simulation experiments. Later on, the number of devices, bandwidth, and communication delay are varied to evaluate the load on the server and the effectiveness of the proposed method in different communication environments.

The settings for the four applications used in the simulation are presented in Tables 5–8. These applications have varying parameters, such as data size and task length, but their operation details have not been specified. The probability of a device using each application is defined as its usage rate. Meanwhile, the usage rate of a virtual machine (VM) is defined as the ratio of the processing capacity required for processing an application.

In this experiment, the probability of offloading from the edge server to the cloud server for each application can be set, and in this case, the probability is uniformly set to 50% for all applications. This parameter only affects one of the baseline methods evaluated in the next section, namely the probability method. There is an equal chance of offloading a task to either the edge server or the cloud server. This approach may be useful in scenarios where the network conditions and server usage are relatively stable and the real-time requirement of tasks is not a significant concern. However, this method may not be optimal in all scenarios.

**Table 2.** Parameters of the cloud server.

| Parameter | Value |
|---|---|
| Number of servers | 1 |
| Number of VMs | 4 |
| Number of cores for VM | 4 |
| MIPS of VM | 10,000 |
| RAM of VM (KB) | 32,000 |
| Storage Capacity of VM (KB) | 1,000,000 |

**Table 3.** Parameters of the edge server.

| Parameter | Value |
|---|---|
| Number of servers | 14 |
| Microprocessor | x86 |
| OS | Linux |
| VMM of Host | Xen |
| Number of cores for Host | 8 |
| MIPS of Host | 4000 |
| RAM of Host | 8000 |
| Storage capacity of Host (KB) | 200,000 |
| Number of VMs | 2 |
| VMM of VM | Xen |
| Number of cores for VM | 2 |
| MIPS of VM | 1000 |
| RAM of VM (KB) | 2000 |
| Storage capacity of VM (KB) | 50,000 |

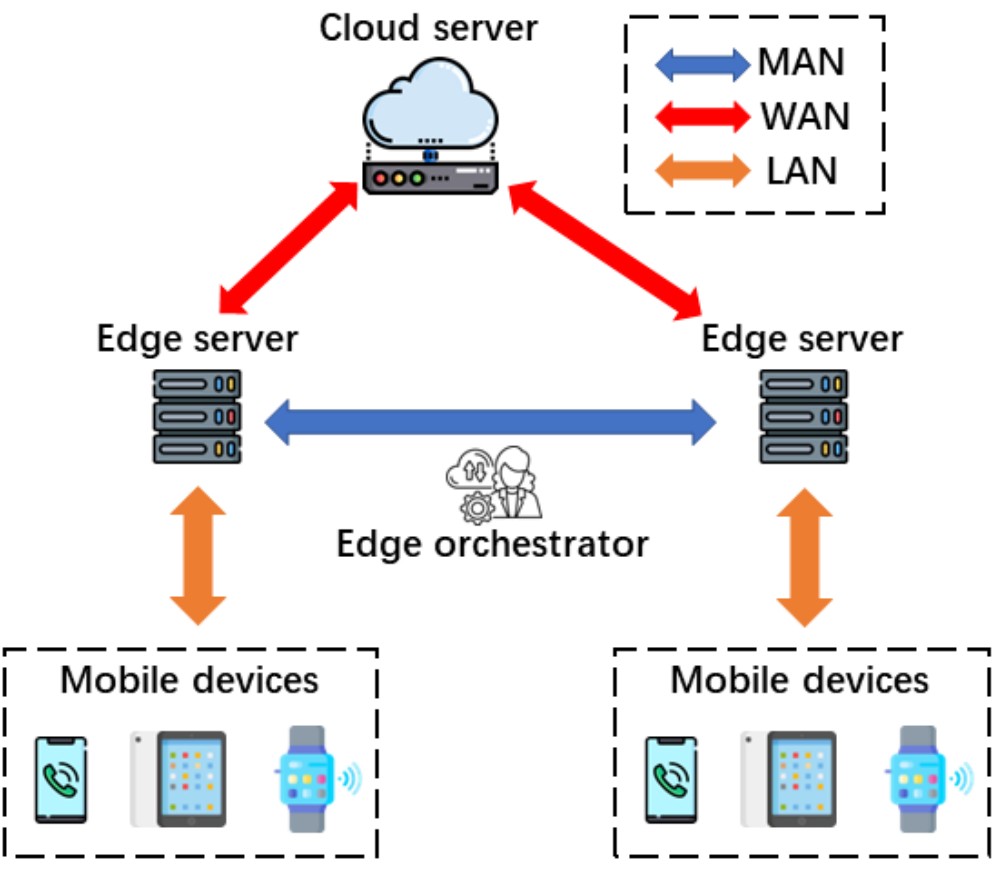

**Figure 2.** Communication environment.

**Table 4.** Simulation parameters.

| Parameter | Value |
|---|---|
| Simulation time (min) | 30 |
| LAN bandwidth (Mbps) | 200 |
| WAN bandwidth (Mbps) | 20 |
| LAN delay (ms) | 1 |
| WAN delay (ms) | 100 |
| The interval of VM load check (s) | 0.1 |
| The interval of VM location check (s) | 0.1 |
| Minimum number of devices | 20 |
| Maximum number of devices | 200 |
| Device count size | 20 |

**Table 5.** Application I.

| Parameter | Value |
|---|---|
| Application usage rate (%) | 20 |
| Probability of offloading to the cloud server (%) | 50 |
| Poisson inter-arrival time (s) | 5 |
| Delay sensitivity | 1 |
| Active period (s) | 45 |
| Idle period (s) | 15 |
| Upload data size (KB) | 150 |
| Download data size (KB) | 25 |
| Task size | 200 |
| Required number of cores | 1 |
| VM utilization on the edge server (%) | 5 |
| VM utilization on the cloud server (%) | 0.5 |

**Table 6.** Application II.

| Parameter | Value |
|---|---|
| Application usage rate (%) | 20 |
| Probability of offloading to the cloud server (%) | 50 |
| Poisson inter-arrival time (s) | 30 |
| Delay sensitivity | 1 |
| Active period (s) | 10 |
| Idle period (s) | 20 |
| Upload data size (KB) | 125 |
| Download data size (KB) | 20 |
| Task size | 400 |
| Required number of cores | 1 |
| VM utilization on the edge server (%) | 5 |
| VM utilization on the cloud server (%) | 0.5 |

**Table 7.** Application III.

| Parameter | Value |
|---|---|
| Application usage rate (%) | 30 |
| Probability of offloading to the cloud server (%) | 50 |
| Poisson inter-arrival time (s) | 60 |
| Delay sensitivity | 0 |
| Active period (s) | 60 |
| Idle period (s) | 60 |
| Upload data size (KB) | 2500 |
| Download data size (KB) | 250 |
| Task size | 3000 |
| Required number of cores | 1 |
| VM utilization on the edge server (%) | 20 |
| VM utilization on the cloud server (%) | 2 |

**Table 8.** Application IV.

| Parameter | Value |
|---|---|
| Application usage rate (%) | 30 |
| Probability of offloading to the cloud server (%) | 50 |
| Poisson inter-arrival time (s) | 7 |
| Delay sensitivity | 0 |
| Active period (s) | 15 |
| Idle period (s) | 45 |
| Upload data size (KB) | 2000 |
| Download data size (KB) | 200 |
| Task size | 200 |
| Required number of cores | 1 |
| VM utilization on the edge server (%) | 10 |
| VM utilization on the cloud server (%) | 1 |

*4.2. Result Analysis*

In the simulation environment described in the previous section, we evaluate and compare the following four methods based on the average service time and task failure rate. The probabilistic method determines the offloading of a task from an edge server to a cloud server based on a preset probability, which remains fixed throughout the simulation as shown in Tables 5–8.

1. Process tasks only on cloud servers (Cloud method).
2. Process tasks only on edge servers (Edge method).
3. Load balancing between edge and cloud servers based on probability (Probability method).
4. Load Balancing between edge and cloud servers using Q-learning (Proposed Method).

The average service time is defined as the sum of the processing time and communication time required to complete a task. The proposed method aims to minimize the average service time. The task failure rate measures the percentage of tasks that fail during processing. In this simulation, there are three types of task failure patterns considered.

1. The user with the device moves while the task is being processed.
2. The task could not be submitted due to network congestion.
3. When the total service time exceeds a threshold.

As the user with the device repeatedly moves over time, there is a possibility that the user may move even before the task is completed, resulting in pattern 1. For patterns 2 and 3, there is an expected waiting time until the task can be sent or processed, but EdgeCloudSim cannot hold the task in both cases. Therefore, even if it falls into pattern 2 or 3, it is considered a task processing failure. If the task failure rate is high, waiting time will typically be generated, increasing the average service time, even if the average service time can be shortened. Thus, this paper evaluates the average service time and task failure rate. Additionally, the change in Q-value is evaluated only in the proposed method.

*4.3. Evaluation in Changing Number of Devices*

In this section, we evaluate how the average service time and task failure rate change with the variation in the number of devices. With an increase in the number of devices, the number of tasks generated also increases, leading to an increase in the load on each server. For this study, we simulate three patterns where the number of devices increases from 20 to 200, 100 to 1000, and 1100 to 2000.

Based on the results presented in Figure 3a,b, it can be observed that the edge method has the longest average service time, while the cloud method has the shortest average service time. Moreover, the task failure rate is less than 1% for all methods, which suggests that waiting for processing on the server and congestion on the network hardly occurred due to the small number of tasks to be processed. These findings indicate that processing with only the cloud server is optimal when the load is small. This is because the task processing time at the cloud server is less than the task communication time.

The change in the Q-value is evaluated in Figure 3c, which shows a gradual increase in the Q-value as the learning progresses. However, occasional decreases in the Q-value can be observed in the second half of the graph, indicating that the number of learning iterations might not be sufficient. Due to a large amount of data, displaying all the Q-values in the graph is impossible. The Q-value is averaged every 200 iterations and presented in Figure 3c.

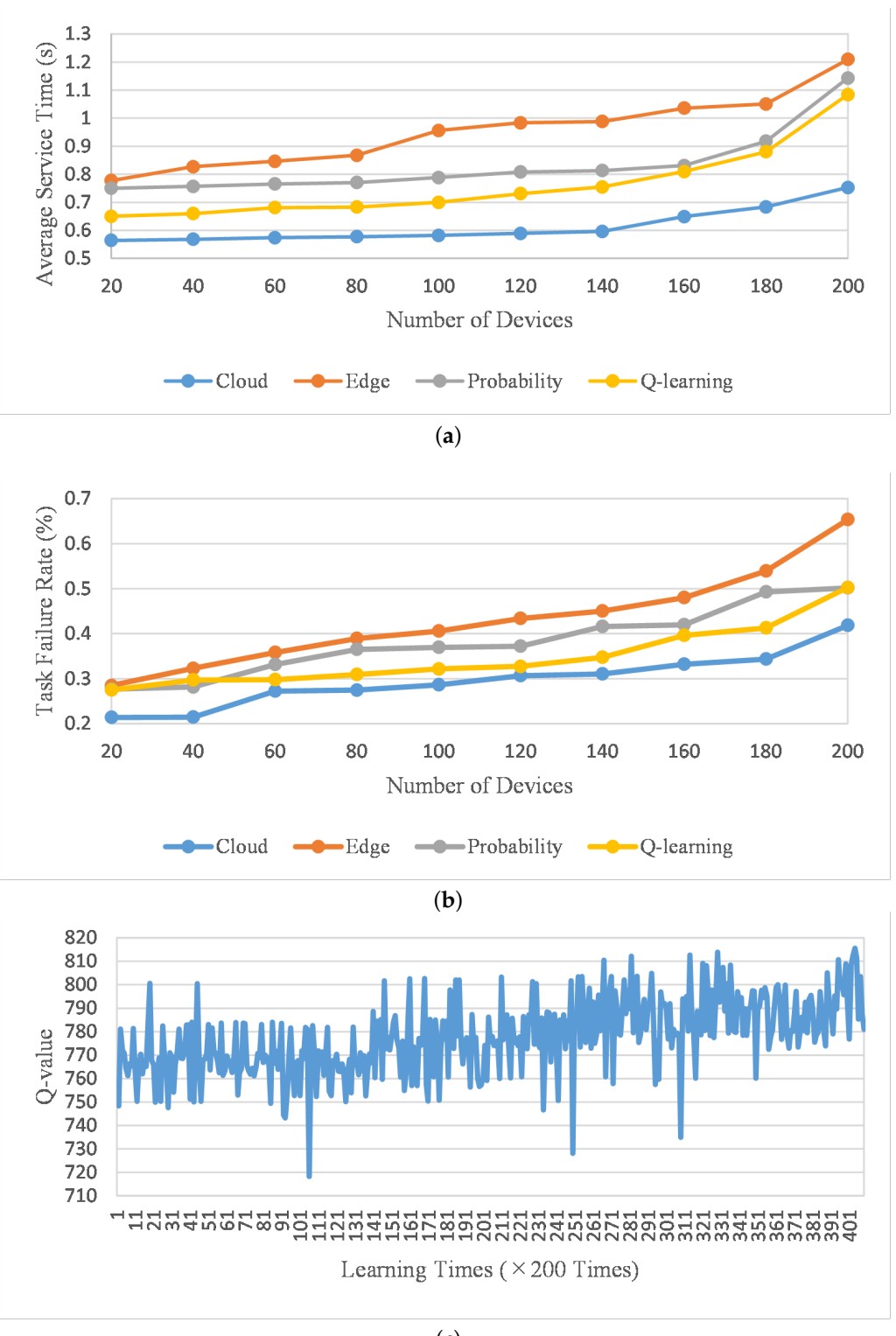

**Figure 3.** When the number of devices is from 20 to 200: (**a**) The comparison of average service time. (**b**) The comparison of task failure rate. (**c**) The change of Q-value.

As shown in Figure 4a, the average service time of the edge method is the longest, and it increases as the number of devices increases. This suggests that it is challenging to reduce the average service time when the load cannot be distributed among the edge servers, as the processing capacity of the edge servers is limited. The proposed method has the shortest average service time, but the cloud method has a shorter average service time when there are fewer than 300 devices. As in the previous section, processing only on the cloud is effective when the load is small, but it becomes difficult as the load increases gradually. As seen in Figure 4b, when the number of devices reaches 1000, the failure rate increases to almost 30%. This is not due to the processing power of the cloud server but due to network congestion preventing the task from being sent successfully. The proposed method shortened the average service time and suppressed the task failure rate to almost 0%. However, since the probabilistic method also suppressed the task failure rate to almost 0%, it is necessary to evaluate where the additional load is applied in the next section.

The variation in Q-value is depicted in Figure 4c. It is evident that the Q-value fluctuates until the number of learning iterations surpasses 301,000 times, after which it stabilizes at a higher value. Increasing the number of learning iterations enables the system to learn diverse scenarios, leading to efficient load distribution. Additionally, as the Q-value increases, it is possible to execute tasks that necessitate real-time processing on the edge server with greater efficiency. The Q-value is averaged every 1000 iterations and presented in Figure 4c.

The results of the average service time are presented in Figure 5a. The proposed method exhibits the shortest average service time when the number of devices is below 1300, but beyond that, the cloud method outperforms it. The average service time decreases as the number of devices increases. In the cloud method, the average service time decreases with increasing load due to the task failure rate. As shown in Figure 5b, the task failure rate surpasses 50% when the number of devices reaches 1300, and it continues to rise, ultimately reaching 70%. The failure rate exceeding 50% implies that less than half of the tasks are processed, resulting in reduced average service times as the load increases. Consequently, in reality, significant waiting time occurs, leading to an enormous average service time, which, in some cases, may even exceed the edge method's time. Moreover, the proposed method reduces the average service time most effectively while considering the task failure rate. Notably, even in the probability method with a 0% failure rate in the preceding section, the failure rate eventually increases to 20%, as demonstrated in the figure 5b. By increasing the network load, the proposed method can reduce the task failure rate more effectively than the probability method. Therefore, the proposed method is demonstrated to execute the most efficient load balancing in heavily loaded server situations.

Figure 5c shows that the Q-value grows by repeating learning as in the previous section and ultimately stabilizes at a higher value. However, despite the number of learning iterations being thrice as high as in the previous section, the Q-value is more varied in the Figure 5c. This variation occurs because the network load surpasses that in the preceding section, leading to a scenario where performing efficient load balancing does not guarantee large rewards. Despite using the proposed method, the task failure rate is nearly 15%, resulting in circumstances where task processing fails, leading to variations in the Q-value. The Q-value is averaged every 3000 iterations and presented in Figure 5c.

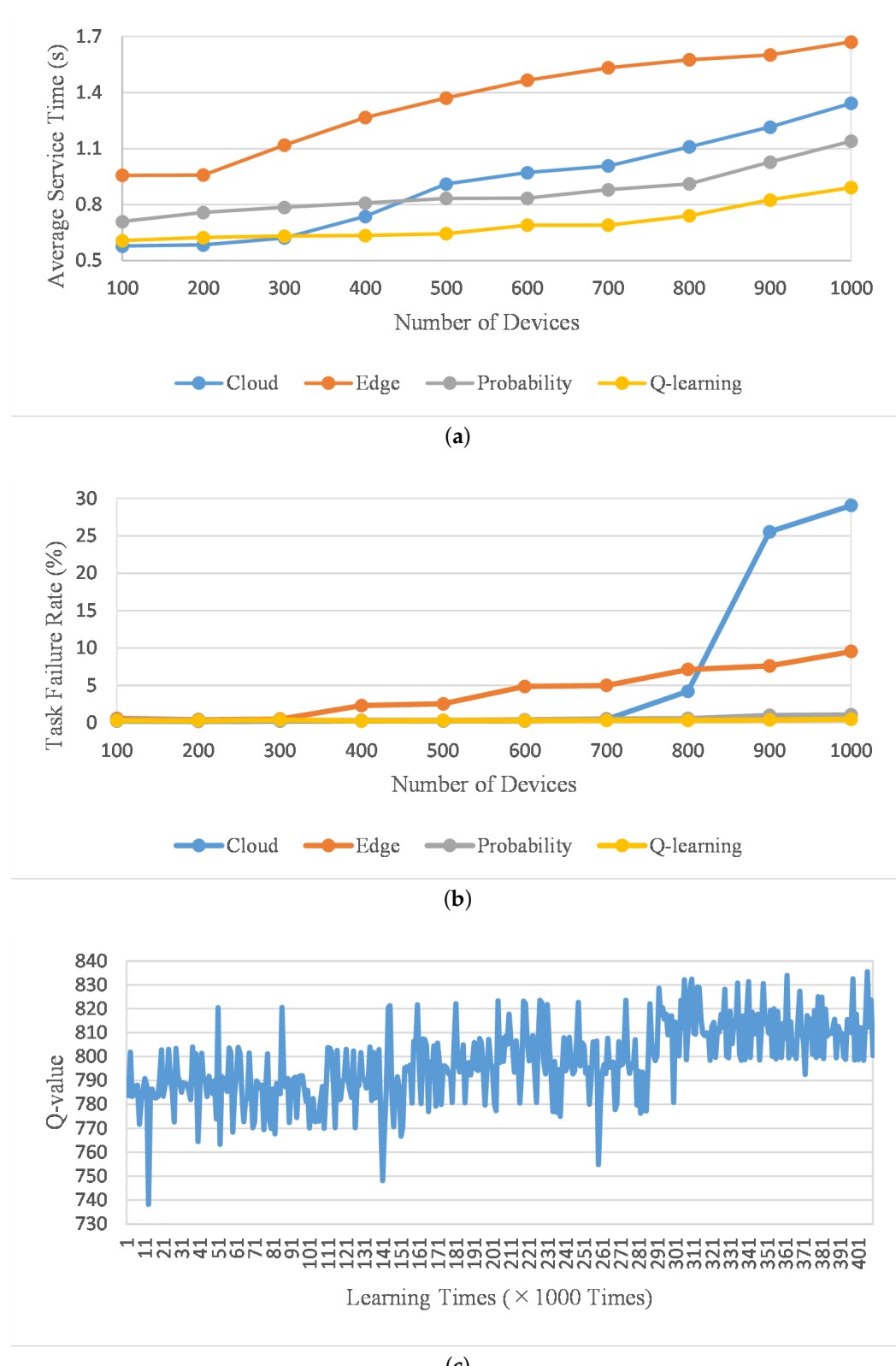

**Figure 4.** When the number of devices is from 100 to 1000: (**a**) The comparison of average service time. (**b**) The comparison of task failure rate. (**c**) The change of Q-value.

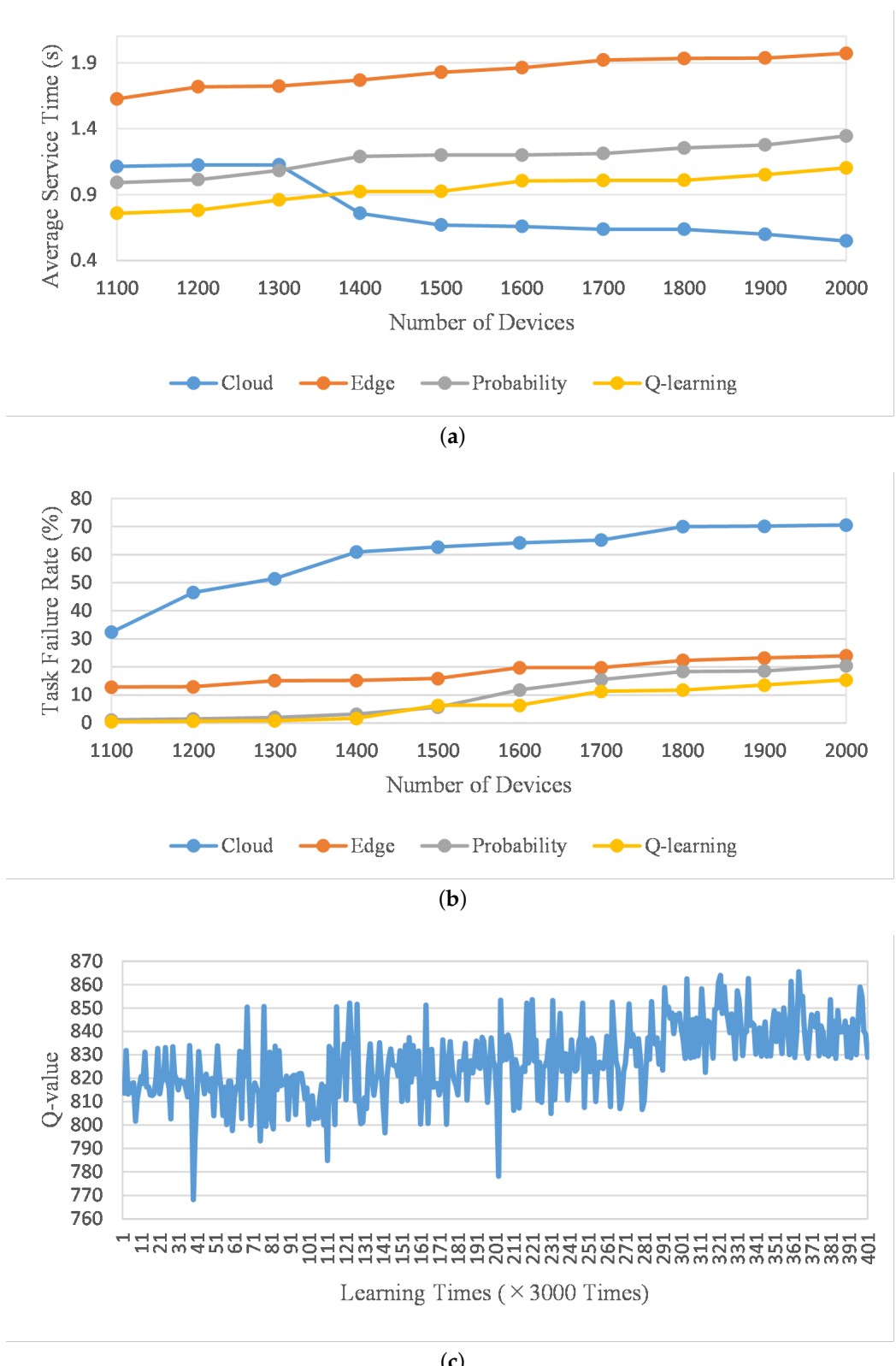

**Figure 5.** When the number of devices is from 1100 to 2000: (**a**) The comparison of average service time. (**b**) The comparison of task failure rate. (**c**) The change of Q-value.

### 4.4. Evaluation in Bandwidth Change

This section assesses the variations in the average service time and task failure rate when the WAN and LAN bandwidths are modified. The measurements are taken when the

number of devices is 1000, with settings for other values remaining unchanged, as shown in the table. The sole difference is in the WAN and LAN bandwidths.

Figure 6 presents the outcome of modifying the WAN bandwidth. Due to the communication environment's configuration, the edge method is insensitive to the WAN bandwidth, thus, neither the average service time nor the task failure rate is affected. Raising the WAN bandwidth for the other three methods can decrease the average service time and the task failure rate. The increased bandwidth augments the number of tasks that can be dispatched, reducing the average service time and decreasing the likelihood of task failures caused by network congestion. Increasing the WAN bandwidth to 30 Mbps yields the cloud method's shortest average service time and almost 0% task failure rate. Thus, the cloud method is optimal if the WAN bandwidth can be increased, whereas the proposed method is the most effective when bandwidth is limited.

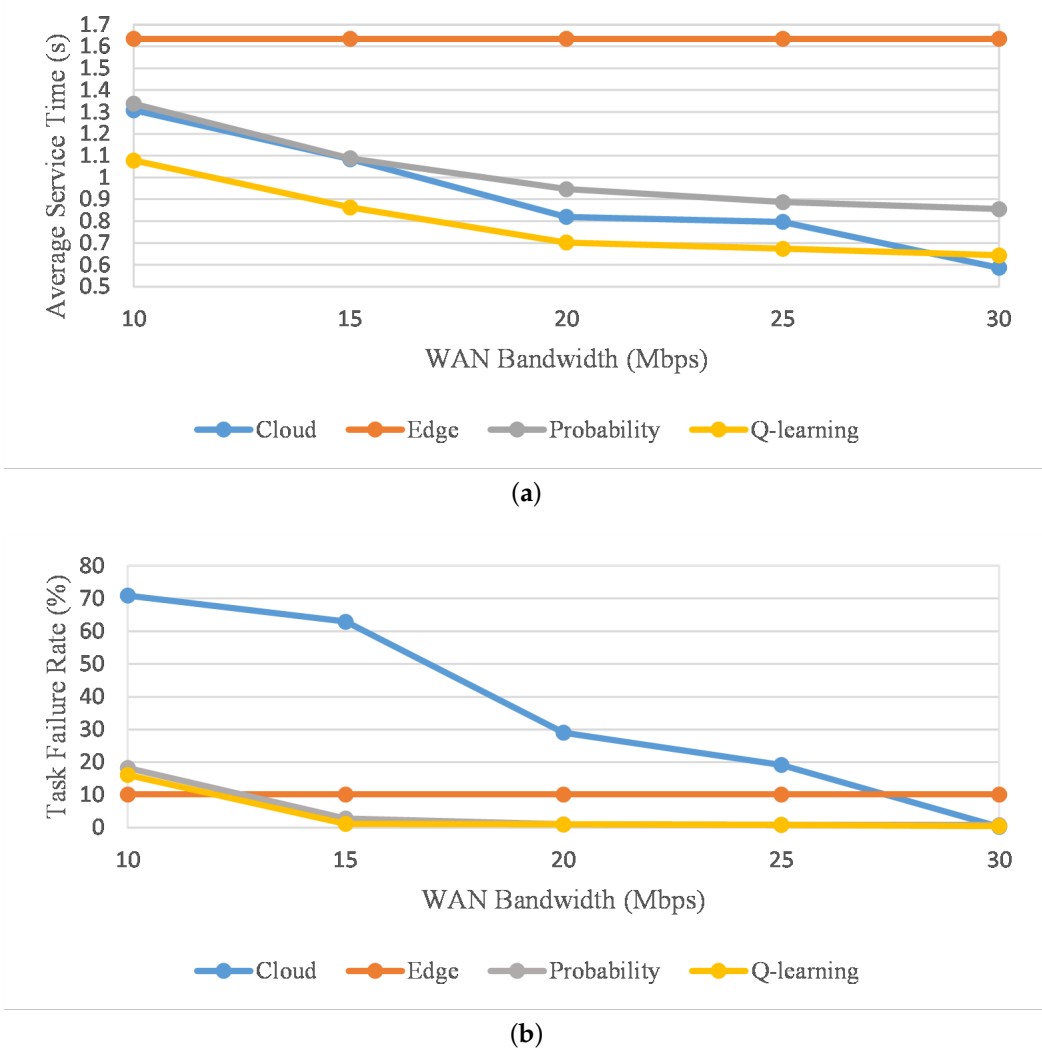

(**a**)

(**b**)

**Figure 6.** Change in WAN bandwidth: (**a**) The comparison of average service time. (**b**) The comparison of task failure rate.

In this section, we present the results of changing the LAN bandwidth, as shown in Figure 7. We observed that increasing the LAN bandwidth can reduce the average service time and task failure rate for all methods. Unlike the original cloud computing, which does not rely on LAN for communication, our proposed method sends tasks to the edge server once, making the cloud method susceptible to changes in LAN bandwidth. Increasing the bandwidth to 250 Mbps resulted in the cloud method's average service time being smaller than that of the probability method. However, since changes in LAN bandwidth

also affect the proposed method, it resulted in the shortest average service time. Therefore, we demonstrate that the proposed method is optimal for average service time and task failure rate, regardless of the LAN bandwidth.

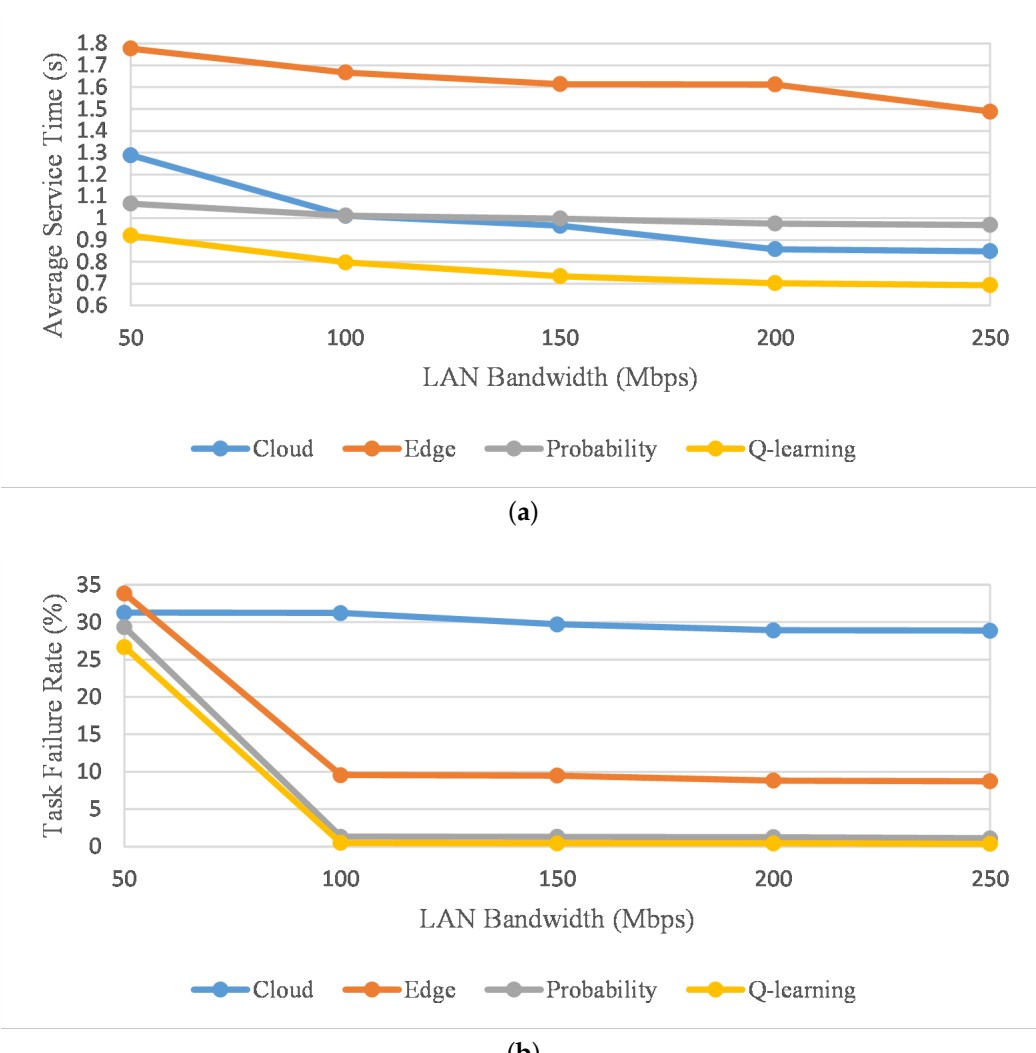

(**a**)

(**b**)

**Figure 7.** Change in LAN bandwidth: (**a**) The comparison of average service time. (**b**) The comparison of task failure rate.

### 4.5. Evaluation of Change in Communication Delay

In this section, we evaluate the impact of WAN and LAN communication delays under the same conditions as the previous section. As shown in Figure 8, similar to WAN bandwidth, the communication delay does not affect edge methods, hence neither the average service time nor task failure rate is affected. However, the other three methods utilize WAN to offload tasks to the cloud server, so longer communication delays resulted in longer average service times. The cloud method experienced an average service time increase of about 0.6 s, the probability method by about 0.5 s, and the proposed method by about 0.4 s. While all methods inevitably increased the average service time, the proposed method proved to be the most effective.

Additionally, as shown in the Figure 8b, the task failure rate decreased as the communication delay time increased. The increased communication delay time extended the interval at which tasks were sent to the cloud server, thus reducing the likelihood of network congestion.

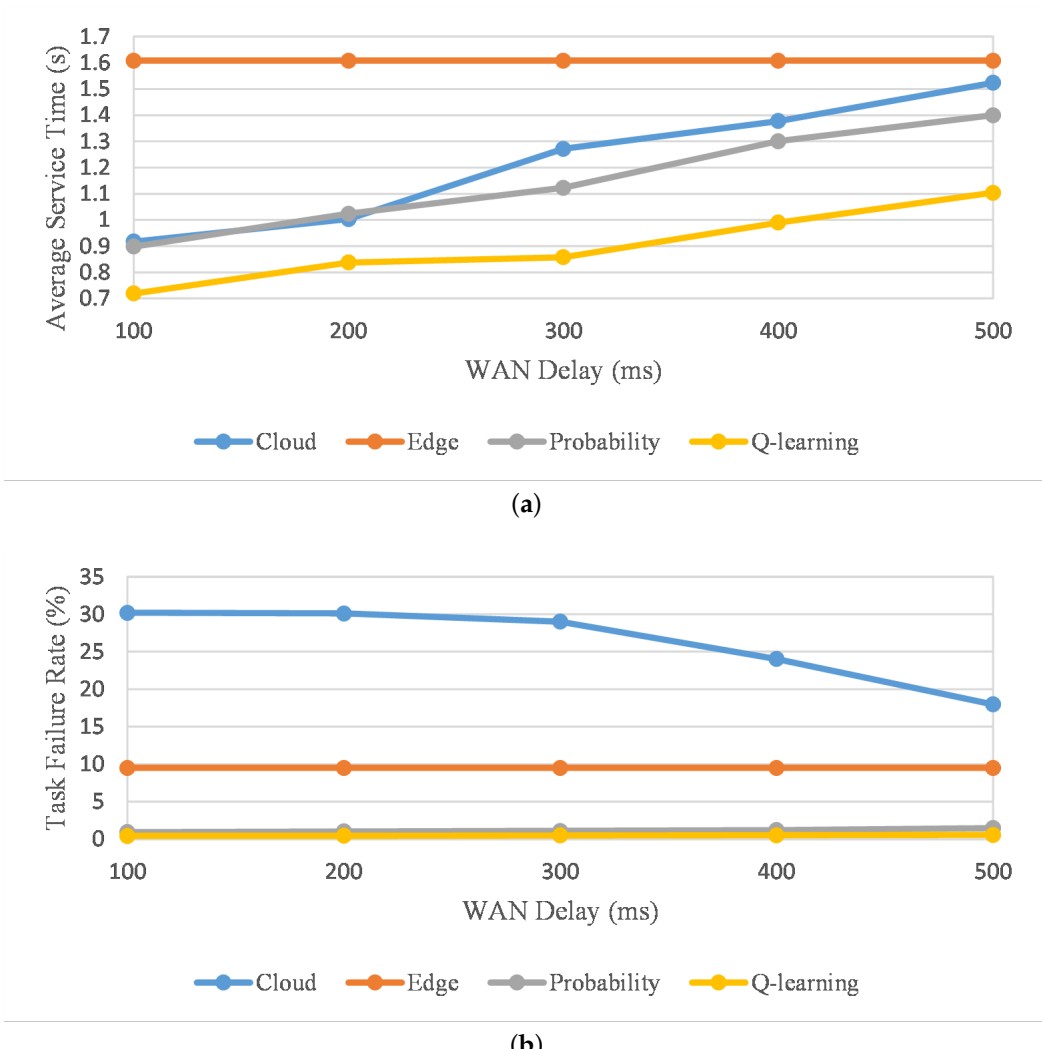

**Figure 8.** WAN communication delay: (**a**) The comparison of average service time. (**b**) The comparison of task failure rate.

As seen from Figure 9, increasing the LAN communication delay increases the average service time for all methods. Since the LAN is always used for communication in any method, it is impossible to avoid an increase in the average service time as the communication delay of the LAN increases. In particular, the edge method, which uses only LAN for communication, has a remarkable effect. However, the increase in average service time for the other three methods is small. The reason for this is that the advantage of using edge computing is that it reduces communication delay and can be used for applications that require real-time performance. Therefore, it is shown that the change in LAN communication delay has almost no effect on average service time and task failure rate. Also, from the Figure 9b, the task failure rate is slightly reduced only for the cloud method. The reason for this is considered to be the same as that mentioned at the end of the previous section.

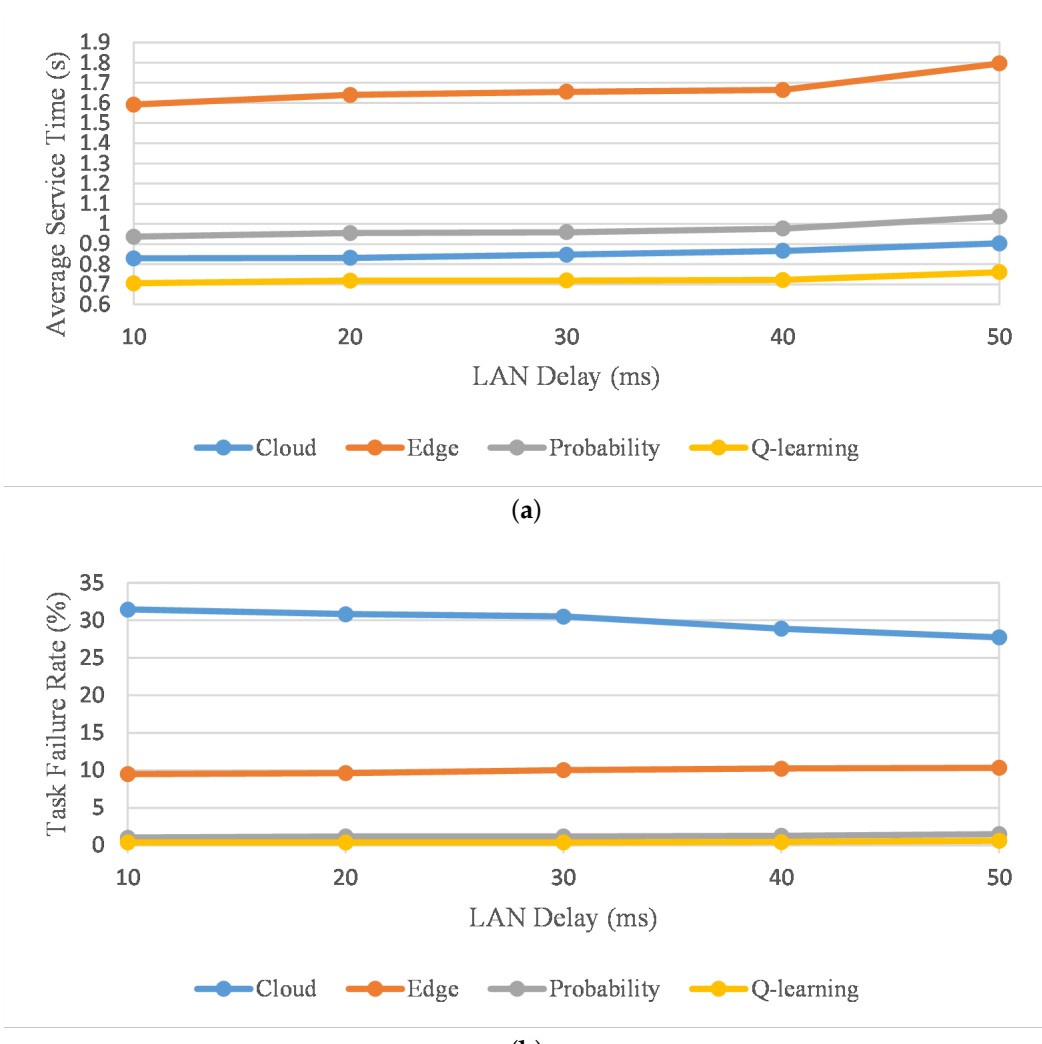

(**a**)

(**b**)

**Figure 9.** LAN communication delay: (**a**) The comparison of average service time. (**b**) The comparison of task failure rate.

## 5. Conclusions

This study introduces a load-balancing method that aims to reduce the average service time of real-time applications by employing reinforcement learning between edge servers and cloud servers. The method utilizes a Q-learning algorithm to dynamically learn the optimal decision-making strategy for task allocation to the most suitable server. This decision-making process takes into account both the real-time requirement of the task and the communication delay between the edge server and the cloud server.

At the core of this approach lies the utilization of a Q-learning algorithm to determine the best action for a given state. The state is defined by three parameters: the task's data size, the processing capacity of the edge server, and the real-time requirement of the task. Based on this information, there are two action options available: processing the task on the edge server or offloading it to the cloud server for processing. The reward is calculated based on the reduction in processing time achieved by selecting a specific action.

Through simulations, we evaluate the proposed method, and the results demonstrate its efficacy in achieving efficient task processing and improving the performance of real-time applications. Moreover, by continuously learning and adapting to the dynamic network conditions and server usage, this method offers an effective solution for load balancing in Edge-Cloud networks. Future research will investigate the potential for further reducing average task service time by incorporating load balancing among different edge servers.

**Author Contributions:** Conceptualization, Z.D. and C.W.; methodology, C.W. and C.P.; software, Z.D.; validation, Z.D. and C.P.; formal analysis, Z.D. and C.P.; investigation, Z.D.; resources, Z.D.; data curation, Z.D.; writing—original draft preparation, Z.D., C.W.; writing—review and editing, C.W., T.Y. and C.P.; visualization, Z.D.; supervision, C.W., C.P. and T.Y.; project administration, C.W.; funding acquisition, C.W. and T.Y. All authors have read and agreed to the published version of the manuscript.

**Funding:** This research is funded by JSPS KAKENHI 22K17880.

**Data Availability Statement:** Not applicable.

**Conflicts of Interest:** The authors declare no conflict of interest.

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
