# Peer review of "A Q-Learning-Based Load Balancing Method for Real-Time Task Processing in Edge-Cloud Networks"

_electronics, doi:10.3390/electronics12153254_

Round 1
Reviewer 1 Report
The title doesn’t seem to be complete. For example, it could be “A Q-Learning Based Load Balancing method/mechanism/algorithm, for Real-Time Task Processing in Edge-Cloud Networks”.
The term “real-time” is not really reflected in the whole work. It seems an important timing variable is missing. After further reading of the work, the actual communication is sequenced rather than concurrent in real time. I would suggest that you can simply remove the word real-time because it has not been tackled. The experimental design of how the Q-learning has been applied within the EdgeCloudSim is not clear or rather skipped. The modelling of the scenarios is not clear as well.
Reviewer 2 Report
The authors describe the load-balancing method for real-time task processing in edge-cloud computing and use simulations to verify the effectiveness of the method.
It seems to me that the formula for Q in the last row of Table 1 is not correctly formulated. The authors do not explain the variables t, a, \gama before using them in the formula. Also, the max function is not paired with proper parentheses to identify the parameters.
Also, at the end of section 3 of the paper, the authors should explain how to determine whether a task should be processed at the edge server or sent to the cloud server for their proposed load balancing method.
In line 230, is there any reason to choose 50%? Do other values change the result significantly?
In line 249, the assumption that a task is counted toward failure when it is queued does not seem to be reasonable. It is better to say a task is counted toward failure when the total service time exceeds a threshold.
In line 5 of Table 1, "Task offload to cloud server" should be "Task offloaded to cloud server".
Reviewer 3 Report
The authors presented the paper "A Q-Learning Based Load Balancing for Real-Time Task Processing in Edge-Cloud Networks." The article submitted is exciting and promising. However, the results presented in the paper are of some value, but their presentation effectively disrupts their reception.
Major observations
- The introduction needs to reformulate the motivation.
- Authors must add a methodology section. This section will give future readers, including graduate students to follow in the footsteps of the presented research.
- In section 4.4, figure 4 (a), the "y" axle numbers need to be replaced because It is difficult to appreciate.
- It is necessary to reformulate the conclusions.
The idea presented and developed in the paper seems promising and exciting. Still, I do not recommend the article for publication due to the above concerns.
- Some sentences may be unclear or hard to follow. Consider rephrasing.
- Some phrases could be wordy. Consider changing the wording.
- The authors should rewrite some sentences to avoid dangling modifier.
- In writing scientific is better to avoid sentences in passive voice because it is easy to follow.
Round 2
Reviewer 2 Report
In the formula for Value Q in the last line of Table 1, the a_{t+1} after max is the variable of the max function. It should be formatted as a subscript of max. It should not be multiplied with Q(S_{t+1}, a_{t+1}).
Reviewer 3 Report
The authors have successfully completed the given observations.
